# Implementation of the Obturator Nerve Block into a Supra-Inguinal Fascia Iliaca Compartment Block Based Analgesia Protocol for Hip Arthroscopy: Retrospective Pre-Post Study

**DOI:** 10.3390/medicina56040150

**Published:** 2020-03-27

**Authors:** Seounghun Lee, Jung-Mo Hwang, Sangmin Lee, Hongsik Eom, Chahyun Oh, Woosuk Chung, Young-Kwon Ko, Wonhyung Lee, Boohwi Hong, Deuk-Soo Hwang

**Affiliations:** 1Department of Anesthesiology and Pain Medicine, Chungnam National University Hospital, Daejeon 34134, Korea; anelee1982@gmail.com (S.L.); lkb333@naver.com (S.L.); tdoreins@naver.com (H.E.); woosuk119@gmail.com (W.C.); annn8432@gmail.com (Y.-K.K.); whlee@cnu.ac.kr (W.L.); 2Department of Orthopedic Surgery, Chungnam National University, Daejeon 34134, Korea; jmtea06@cnuh.co.kr; 3Department of Anesthesiology and Pain Medicine, College of Medicine, Chungnam National University, Daejeon 34134, Korea; 5chahyun@naver.com; 4Department of Orthopedic Surgery, College of Medicine, Chungnam National University, Daejeon 34134, Korea

**Keywords:** hip arthroscopy, supra-inguinal fascia iliaca compartment block (SI-FICB), obturator nerve block, analgesic effect

## Abstract

*Background and Objectives:* The effect of supra-inguinal fascia iliaca compartment block (SI-FICB) in hip arthroscopy is not apparent. It is also controversial whether SI-FICB can block the obturator nerve, which may affect postoperative analgesia after hip arthroscopy. We compared analgesic effects before and after the implementation of obturator nerve block into SI-FICB for hip arthroscopy. *Materials and Methods:* We retrospectively reviewed medical records of 90 consecutive patients who underwent hip arthroscopy from January 2017 to August 2019. Since August 2018, the analgesic protocol was changed from SI-FICB to SI-FICB with obturator nerve block. According to the analgesic regimen, patients were categorized as group N (no blockade), group F (SI-FICB only), and group FO (SI-FICB with obturator nerve block). Primary outcome was the cumulative opioid consumption at 24 hours after surgery. Additionally, cumulative opioid consumption at 6 and 12 hours after surgery, pain score, additional analgesic requests, intraoperative opioid consumption and hemodynamic stability, and postoperative nausea and vomiting were assessed. *Results:* Among 87 patients, there were 47 patients in group N, 21 in group F, and 19 in group FO. The cumulative opioid (fentanyl) consumption at 24 hours after surgery was significantly lower in the group FO compared with the group N (N: 678.5 (444.0–890.0) µg; FO: 482.8 (305.8–635.0) µg; *p* = 0.014), whereas the group F did not show a significant difference (F: 636.0 (426.8–803.0) µg). *Conclusion*: Our findings suggest that implementing obturator nerve block into SI-FICB can reduce postoperative opioid consumption in hip arthroscopy.

## 1. Introduction

Cases of hip arthroscopy have been increasing in various diseases of the hip joint, such as femoro-acetabular impingement, osteochondritis, labral tear, removal of loose bodies, septic arthritis, or unknown hip pain [1]. Despite the minimally invasive nature of arthroscopic surgery compared to open surgery, hip arthroscopy accompanies severe postoperative pain in most cases [2,3]. Therefore, adequate analgesia is important in hip arthroscopy, not only for pain relief, but because it is also associated with faster recovery, effective rehabilitation, increased patient satisfaction, and early discharge [4,5]. Various analgesic techniques have been suggested, of which peripheral nerve blocks are known to be effective [5,6,7]. 

The hip joint is innervated by the branches of the lumbar (femoral, obturator nerve) and sacral plexus (quadratus femoris, superior gluteal, sciatic nerve) [8]. Lumbar plexus block produces anesthesia of the anterior part of the hip capsule where the greatest concentration of sensory nerve endings and mechanoreceptors are found [9]. It has been suggested as an effective analgesic technique after hip arthroscopy [7,10]. One of the anterior approach to the lumbar plexus block is supra-inguinal fascia iliaca compartment block (SI-FICB) [11,12]. This technique, theoretically, can block the femoral, obturator, and lateral femoral cutaneous nerves simultaneously, since these nerves are contained within the same compartment [13]. 

However, reliable involvement of the obturator nerve by FICB is rather controversial [12,14]. According to Bendtsen et al. [15], the spread of local anesthetic to the obturator nerve can hardly occur after SI-FICB because of its medial and posterior location relative to the psoas muscle and separation from the compartment at the level of the lesser pelvis. If SI-FICB cannot reliably provide blockade of the obturator nerve, its analgesic effect can be limited by the sparing of the anteromedial side of the hip joint, which is innervated by the articular branch of the obturator nerve [16]. 

We initially introduced SI-FICB into our clinical practice as a component of multimodal analgesia after hip arthroscopy, but its effect was not as marked as we expected. As several articles questioning the analgesic effect of SI-FICB for hip arthroscopy have been published [17,18], we changed our protocol to supplementing obturator nerve block to the SI-FICB. This study retrospectively analyzed postoperative analgesic outcomes by comparing opioid consumption according to the change of analgesic protocol.

## 2. Materials and Methods

This retrospective pre-post study was approved by the Institutional Review Board of Chungnam National University Hospital (2019-07-050), and the trial was registered at a clinical trial registry (KCT0004419). Informed consent was waived, due to the retrospective design of the study using medical records. This article adheres to the applicable Strengthening the Reporting of Observational Studies in Epidemiology (STROBE) guidelines [19].

### 2.1. Implementation of Subpectineal Obturator Nerve Block

SI-FICB for hip arthroscopy has been performed by a single anesthesiologist (B.H.) since 2017 in our institution. In the meantime, it was recognized that the pain score or opioid consumption were independent of SI-FICB in hip arthroscopy patients. It was thought this was due to the lack of coverage of SI-FICB for the complex innervation of hip joints. Since the involvement of obturator nerve by FICB has been an issue [14], we thought that supplemental obturator nerve block after SI-FICB might be helpful for effective analgesia. Thereafter, supplemental obturator nerve block was implemented with the existing multimodal analgesia protocol from August 2018 for better postoperative analgesia including opioid consumption [20]. Patients who had not had any nerve blocks during the same period were assigned to group N as the control group. Thus, the patients were divided into three groups: group N (no blockade), group F (SI-FICB only), and group FO (SI-FICB with obturator nerve block) (Figure 1).

### 2.2. Data Collection

The electronic medical records of patients who underwent hip arthroscopy performed by a single experienced surgeon (D.H.) from January 2017 to August 2019 in Chungnam National University Hospital were reviewed retrospectively. Patients with chronic opioid use, inadequate medical records as regards the analgesic outcome, and American Society of Anesthesiologists physical status classification above 3 were excluded. The medical records were collected and reviewed by a trained coordinator (Sh.L.) using a standardized form, and an independent investigator (B.H.) analyzed the data after de-identification.

### 2.3. Supra-Inguinal Fascia Iliaca Compartment Block and Subpectineal Obturator Nerve Block

The nerve blockade was performed by a single anesthesiologist (B.H.) immediately before the induction of general anesthesia under ultrasound guidance using an in-plane technique with Mylab^TM^ 25 Gold (Esaote, Genova, Italy) and a linear probe (LA435: 6–18 MHz, Esaote, Genova, Italy). A 22 gauge, 100 mm, echogenic needle (SonoPlex cannulas, Pajunk®, Geisingen, Germany) was used. Povidone-iodine was used to make an aseptic field. 

For SI-FICB, the method was standardized as described by Hebbard et al. [11]. The probe was placed over the inguinal ligament. After checking the common femoral artery and femoral nerve, the lateral portion of the femoral nerve was centered on the screen. Then the lateral part of the probe was rotated cranially to capture the parasagittal plane. The deep circumflex iliac artery was used as a landmark for needle placement, which was located superficial to the fascia iliaca, 1–2 cm superior to the inguinal ligament. The needle was introduced in the caudal-to-cranial direction until the tip of the needle was positioned under the fascia iliaca at the level of the deep circumflex iliac artery. After confirming successful hydro-dissection between the fascia iliaca and the iliaca muscle using 1–2 mL of local anesthetic, in total 40 mL of 0.25% ropivacaine was injected in several aliquots. To facilitate the spread of local anesthetic to the proximal lumbar plexus, the needle was passed cranially between the injections, deep to the fascia iliaca and into the iliac fossa, moving only into the space created by the distending fluid. The short-axis view was also reviewed during the injection to see the proper spread of local anesthetic around the femoral nerve by rotating the probe parallel to the inguinal ligament. 

The subpectineal approach described by Taha et al. was used to block the articular branch of the obturator nerve [21]. The medial inguinal region was scanned to identify the pectineus muscle. Then the probe was tilted cranially and traced the muscle until the superior pubic ramus was visualized. The block needle was advanced toward the most medial part of the fascia separating the pectineus and external obturator muscle, and 20 mL of 0.25% ropivacaine was injected to achieve spread of local anesthetic within the intermuscular fascial plane deep to the pectineus muscle.

### 2.4. Anesthesia and Surgery

Patients were premedicated with anticholinergics (0.04 mg/kg of glycopyrrolate). Routine monitoring included electrocardiography, pulse oximetry, and non-invasive blood pressure measurements. Then general anesthesia was performed by standard methods using 1.5 mg/kg of propofol, 0.8 mg/kg of rocuronium, followed by tracheal intubation and maintained with inhaled desflurane and 0.05–0.2 µg/kg/min of remifentanil. Hip arthroscopy was carried out using the lateral approach in a supine position and the standard 3- to 4-portal technique in all cases.

One gram of paracetamol and 0.5 µg/kg of fentanyl were administered at the end of surgery. The patient-controlled analgesia (PCA) device (GemStar^TM^, Hospira, Lake Forest, IL, USA) was programmed to administer a bolus dose of 0.5 µg/kg of fentanyl, with a lockout time of 10 min, without background infusion. One hundred milliliters of solution containing 1500 µg of fentanyl and 0.6 mg of ramosetron was used for PCA. It is our routine to explain the numeric rating scale (NRS) (0, no pain; 10, worst pain), when PCA should be used (NRS ≥ 4 or when the patient feels pain), and how to use the PCA device while filling out the consent form for anesthesia on the day before surgery. If the pain was not controlled (NRS ≥ 4) by the PCA, 25 mg of pethidine was administered and repeated if necessary. All patients received 20 mg of nefopam intravenously and 650 mg of oral acetaminophen every 8 hours postoperatively as a part of multimodal analgesia.

### 2.5. Outcome Measures

The primary outcome was cumulative opioid consumption at 24 hours after surgery. The cumulative opioid consumption at 6 and 12 hours after surgery, pain score, additional analgesic requests, intraoperative opioid consumption and hemodynamic stability, and the incidence of postoperative nausea and vomiting were assessed as secondary outcomes. Numeric rating scale (NRS) pain score was assessed in the post-anesthesia care unit and on the ward by regular nurse rounds and patient requests for pain control. Intraoperative hemodynamic stability was assessed by the incidence of hypertension (systolic blood pressure above 140 mmHg) or hypotension (systolic blood pressure below 90 mmHg) and the need for vasoactive drugs (0.5 mg of nicardipine or 50 µg of phenylephrine). 

### 2.6. Sample Size Calculation and Statistical Analysis

The study sample size was chosen as all the patients undergoing hip arthroscopy from January 2017 to August 2019. All available patients were considered, and priori power analysis was conducted. According to our previous unpublished data, the cumulative consumption of fentanyl at 24 hours after hip arthroscopy was about 700 µg (SD 150 µg). We determined that the 30% reduction of fentanyl consumption has a clinical effect (effect size 1.4). Based on the difference between two independent means by t-test, 13 patients per group were required to give a power of 80% with an alpha of 0.016 (0.05/3) for post hoc analysis when three groups are significant. Considering the number of annual cases (30 cases average), and the ratio of utilizing peripheral nerve block-based multimodal analgesia (about a 1:1 ratio), this figure was expected to be more than 13 per group.

All statistical analyses were performed using R software (version 3.6.1, R Project for Statistical Computing, Vienna, Austria). The normality of continuous data was assessed using the Shapiro–Wilk test. The homogeneity of variances was assessed using Bartlett’s test. If both conditions were satisfied, comparisons between groups were determined by analysis of variance (ANOVA), with the results expressed as mean ± SD. If homogeneity of variance was unmet, Welch’s ANOVA was performed. If normality was not satisfied, groups were compared using the Kruskal–Wallis test, with the results expressed as median (IQR). The pairwise comparisons for post hoc analysis were analyzed using Tukey’s honestly significant difference test after ANOVA and Welch‘s ANOVA, Dunn’s test after Kruskal–Wallis test with p-value after adjustment for the multiple comparisons. Categorical data were compared using the χ^2^ test or Fisher’s exact test, as appropriate, with the results expressed as number (%). A two-tailed *p*-value < 0.05 was considered statistically significant. We used the function of the R program using conditional statements for iterative and automated statistical processing. Statistical results of perioperative opioid consumptions are presented as Appendix A.

## 3. Results

A total of 90 patients were screened for study participation. Of those, 3 patients were excluded (1 inadequate medical record, 1 chronic opioid user, 1 did not receive surgery after general anesthesia due to hemodynamic instability). Consequently, a total of 87 patients were enrolled in the study, and 21 patients received SI-FICB (group F, 19 months pre) while 19 patients received SI-FICB with obturator nerve block (group FO, 13 months post). Forty-seven patients did not receive any nerve block during the study period (group N). The baseline patient characteristics did not differ significantly among the groups (Table 1). 

The perioperative opioid use including primary outcome is presented in Table 2. Ten patients (2 in group F, 3 in group FO, 5 in group N) were excluded from the comparison of opioid consumption because of lost data regarding PCA use. The cumulative opioid (fentanyl) consumption at 24 hours after surgery was significantly lower in group FO compared with group N (N: 678.5 (444.0–890.0) µg; FO: 482.8 (305.8–635.0) µg; *p* = 0.014), whereas group F did not show a significant difference (F: 636.0 (426.8–803.0)) (Table 2). The cumulative opioid consumption at 6 and 12 hours after surgery was significantly lower in group FO compared to group N, whereas group F did not differ from group N. The intraoperative remifentanil doses were significantly lower in groups F and FO. The plot of postoperative opioid consumptions are presented as Appendix A (Appendix A).

Secondary outcomes are presented in Table 3. Despite the fact that postoperative opioid consumption for 24 hours was lowest in the FO group, the incidence of postoperative nausea and vomiting (PONV) showed no difference. The pain scores in post-anesthesia care unit (PACU) and the lowest pain scores during the 24 hours after surgery were significantly different among the three groups, but the highest pain scores were not different. There was no difference in the indicators of intraoperative hemodynamic instability. 

## 4. Discussion

The present study demonstrates that SI-FICB with obturator nerve block improves postoperative analgesia after hip arthroscopy. SI-FICB alone was not able to show a significant difference in opioid consumption compared with the control group. This indicates that involvement of the obturator nerve is crucial in effective analgesia after hip arthroscopy. At the same time, this suggested that SI-FICB cannot provide reliable blockade of the obturator nerve.

There are multiple nerves innervating the hip joint [8]. Each antero-lateral, antero-medial, postero-lateral, and postero-superior area of the hip joint is innervated by femoral, obturator, superior gluteal, and sciatic nerves. Because of this complex innervation of the hip joint, various nerve-blocking techniques have been used for postoperative analgesia after hip arthroscopy [6]. However, to date none of these techniques are considered as gold standard [22]. 

Among several regional analgesic techniques, SI-FICB significantly reduces opioid consumption after total hip arthroplasty [23] or hip fractures [24], unlike the classical infra-inguinal approach [25]. When the two approaches were compared directly in total hip arthroplasty, SI-FICB showed superior analgesic efficacy [26]. The authors of these studies explained their results with the theory that SI-FICB has better cephalic spread of local anesthetic than the infra-inguinal approach, which may result in blockade of all three nerves of the lumbar plexus. This theory, however, does not fit with our results.

On the other hand, the analgesic effect of SI-FICB in hip arthroscopy was not manifested apparently in previous studies [18,27,28]. Alrayashi et al. [27] reported lesser perioperative opioid consumption after SI-FICB in adolescent and young adult patients; however, there was no difference in pain score in PACU, and about one-third of the patients suffered severe pain regardless of the blockade. Eastburn et al. [28] also reported moderate or severe pain in about half of the patients (8/17 patients) despite what the authors concluded was successful blockade in 94% of cases. This discrepancy in the analgesic effect of SI-FICB between hip arthroplasty and arthroscopy is interesting. It is demonstrated in the current study that supplemental obturator nerve block can make a significant difference to the analgesic outcome of SI-FICB after hip arthroscopy. This result may offer a clue as to where the discrepancy resides.

It is not clear whether the blockade of obturator nerve had occurred in previous studies using SI-FICB [18,23,26,27,28]. The sensory distribution of the obturator nerve shows a lot of variation over the medial thigh [29], and the femoral nerve also contributes to hip adduction. Therefore, physical assessment of successful blockade of the obturator nerve can be less clear when FICB is performed. Also, when motor function is assessed postoperatively, it can be complicated by post-surgical pain [23].

There are several limitations in our study. First, there may have been some hidden confounding due to the retrospective study design. Second, the effect size was not enough to permit direct comparison of the combined blockade (SI-FICB with obturator nerve blockade) and SI-FICB. Third, we cannot conclusively state that SI-FICB per se cannot involve obturator nerve block. Confirmatory imaging or anatomical research is needed. Finally, despite the fact that the combination used in this study showed significant reduction in opioid consumption, it was not possible to provide sufficient analgesia in all cases. We expect more promising results from future studies using other methods to block periarticular branches [30], such as the recently described pericapsular nerve group (PENG) block or iliopsoas plane block [31,32].

## 5. Conclusions

SI-FICB with obturator nerve block can reduce opioid consumption in hip arthroscopy. Our study suggests that SI-FICB cannot reliably block obturator nerve. Further prospective studies without confounding factors and specific anatomical studies are warranted to alleviate postoperative pain after hip arthroscopy.

## Figures and Tables

**Figure 1 medicina-56-00150-f001:**
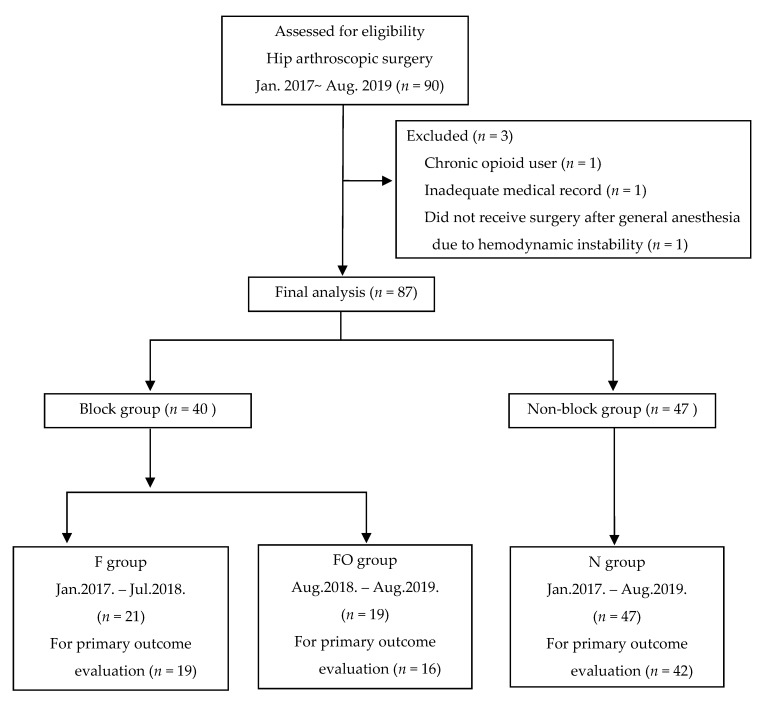
Flow diagram of participants through the study. F, fascia iliaca compartment block group; FO, fascia iliaca compartment block and obturator nerve block group; N, non-block group.

**Table 1 medicina-56-00150-t001:** Demographic and perioperative characteristics of patients.

	F (N = 21)	FO (N = 19)	N (N = 47)
Sex			
- Female	10 (47.6%)	10 (52.6%)	22 (46.8%)
- Male	11 (52.4%)	9 (47.4%)	25 (53.2%)
Age	37.0 (27.0;52.0)	38.0 (25.5;45.0)	39.0 (26.0;47.0)
Weight (kg)	69.2 (61.2;77.0)	67.9 (63.1;72.3)	67.3 (57.6;77.6)
Height (cm)	166.0 (161.4;173.6)	166.6 (161.0;171.2)	167.1 (162.8;176.0)
BMI (kg/m^2^)	24.7 (23.3;26.8)	24.8 (23.1;26.7)	23.7 (21.8;25.8)
Anesthesia time (min)	165.7 ± 29.2	184.2 ± 29.9	178.0 ± 26.8
Operation time (min)	138.0 ± 28.6	158.1 ± 27.6	156.5 ± 27.6
Hypertension	1 (4.8%)	2 (10.5%)	7 (14.9%)

Data are presented as the number (%); median (interquartile range); or as the mean ± standard deviation (SD). BMI, body mass index; F, SI-FICB only; FO, SI-FICB with obturator nerve block; N, no blockade.

**Table 2 medicina-56-00150-t002:** Perioperative opioid consumption including primary outcome.

	F (N = 19)	FO (N = 16)	N (N = 42)	Overall p	FO vs. F (p adj.)	FO vs. N (p adj.)	F vs. N (p. adj.)
Cumulative postoperative fentanyl consumption (µg)							
6 hours	249.6 (161.3;350.3)	163.0 (108.0;341.5]	293.0 (210.6;385.0)	0.028	0.256	0.028	0.258
12 hours	372.6 (297.0;458.0)	256.2 (186.1;420.0]	402.0 (300.0;582.0)	0.024	0.207	0.020	0.289
24 hours	636.0 (426.8;803.0)	482.8 (305.8;635.0]	678.5 (444.0;890.0)	0.018	0.066	0.014	0.598
	**F (N = 21)**	**FO (N = 19)**	**N (N = 47)**				
Intraoperative remifentanil dose (µg/kg/min)	0.069 (0.056;0.075)	0.059 (0.055;0.068]	0.089 (0.069;0.102)	0.001	0.320	0.003	0.004

Data are presented as median (interquartile range). p adj., p value adjustment.

**Table 3 medicina-56-00150-t003:** Comparison of secondary outcomes between groups.

	F (N = 21)	FO (N = 19)	N (N = 47)	p
Pain score during postoperative 24 hours				
PACU	4.0 (3.0;5.0)	3.0 (2.0;4.0)	4.0 (3.0;6.0)	0.039
Lowest	2.0 (2.0;2.0)	2.0 (1.0;2.0)	2.0 (2.0;2.5)	0.039
Highest	4.0 (3.0;6.0)	5.0 (3.5;5.5)	5.0 (4.0;7.0)	0.124
Additional Demerol usage	4 (19.0%)	7 (36.8%)	15 (31.9%)	0.426
Hypertension	2 (9.5%)	0 (0.0%)	2 (4.3%)	0.352
Hypotension	7 (33.3%)	10 (52.6%)	16 (34.0%)	0.327
Phenylephrine (0/1/2/3)	14/5/1/1	9/7/3/0	32/9/4/2	0.551
Nicardipine (0/1/2/3)	19/2/0/0	17/2/0/0	43/1/1/2	0.556
PONV	7 (33.3%)	6 (31.6%)	14 (29.8%)	0.957

Data are presented as the number (%); median (interquartile range); PACU, post-anesthesia care unit; PONV, postoperative nausea and vomiting.

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
