# Peer review of "Implementation of the Obturator Nerve Block into a Supra-Inguinal Fascia Iliaca Compartment Block Based Analgesia Protocol for Hip Arthroscopy: Retrospective Pre-Post Study"

_medicina, 2020, doi:10.3390/medicina56040150_

Round 1
Reviewer 1 Report
The submission addresses a very interesting topic of anesthesia protocol suitable for hip arthroscopy.
The missing information is the pain threshold influence on the fentanyl consumption
The authors must explain for what test thirteen patients per group have a power of 80% with an alpha of 0.016.
BY definition the ANOVA testing requires normal distribution. There is no proof in the tables or text showing the normal distribution of variables.
There is a huge difference when the Pain score is assessed in a range of 3 hours. The authors wrote that assessment was performed every 3~6 hours in a numeric rating scale (NRS) during the stay in the post-anesthesia care unit and ward.
The authors applied different protocols for anesthesia. How did they keep the randomization protocol for this study?
What kind of informed consent was signed by patients?
How much the study is biased while choosing SI-FICB with or without an obturator nerve block?
How the informed consent could be waived if the patients were divided into three groups: group N (no blockade), group F (SI-FICB only), and group FO (SI-FICB with obturator nerve block). How the protocol was presented to IRB and how the approval was obtained?
The fentanyl use in the postoperative period should be clearly described how its consumption changes, and sources and types of influences.
Author Response
Thank you for giving me the opportunity to review the manuscript titled: “Implementation of the obturator nerve block into a supra-inguinal fascia iliaca compartment block based analgesia protocol for hip arthroscopy: Retrospective Pre-Post study” written by Seounghun Lee et al.
Point 1: The submission addresses a very interesting topic of anesthesia protocol suitable for hip arthroscopy. The missing information is the pain threshold influence on the fentanyl consumption
We have modified the manuscript as adviced.
‘It is our routine to explain the NRS (0, no pain; 10, worst pain), when PCA should be use (NRS ≥ 4 or when patient feels pain) and how to use the PCA device while filling out the consent form for anesthesia on the day before surgery’
The authors must explain for what test thirteen patients per group have a power of 80% with an alpha of 0.016.
We thank the reviewer for the comment. We have modified the manuscript as adviced.
‘Based on difference between two independent means by t-test, thirteen patients per group were required to have a power of 80% with alpha of 0.016 (0.05/3) for post hoc analysis when three groups are significant.’
BY definition the ANOVA testing requires normal distribution.
There is no proof in the tables or text showing the normal distribution of variables.
Yes. You are right. Our main result showed by median [IQR]. The groups were compared using the Kruskal-Wallis test according to Shapiro Wilk-test. We described in the statistical session.
There is a huge difference when the Pain score is assessed in a range of 3 hours. The authors wrote that assessment was performed every 3~6 hours in a numeric rating scale (NRS) during the stay in the post-anesthesia care unit and ward.
We have modified the manuscript as follow.
‘Numeric rating scale (NRS) pain score was assessed in post-anesthesia care unit and in ward by nurse regular rounding and patient request for pain control.’
So, we expressed only highest and lowest NRS during postoperative 24 hours.
The authors applied different protocols for anesthesia. How did they keep the randomization protocol for this study?
What kind of informed consent was signed by patients?
How much the study is biased while choosing SI-FICB with or without an obturator nerve block?
How the informed consent could be waived if the patients were divided into three groups: group N (no blockade), group F (SI-FICB only), and group FO (SI-FICB with obturator nerve block). How the protocol was presented to IRB and how the approval was obtained?
As stated in the title, this is not RCT. We restrospectively collected and analyzed the data.
The background of our hip arthroscopy postoperative analgesia protocol changes is detailed in the 2. 1 Implementation of subpectineal obturator nerve block.
The fentanyl use in the postoperative period should be clearly described how its consumption changes, and sources and types of influences.
I think the meaning is that it was possible to reduce opioid consumption in FO group.
I'm really sorry, but if the question is not answered properly, please provide me with directions to supplement it in more detail.

Reviewer 2 Report
This article is very well written and designed. The designed purpose is legible and the comprehensive investigation is identified effectively.
The authors are descriptive, identifying the results in a clear and concise way.
The authors also identify areas for furthered research and limitations for current results.
On a grammatical scale, the article is articulate and exact.
Author Response
Special thanks to you for your good comments on our article.
Hopefully, this article provides instructive points for other healthcare professionals.
Round 2
Reviewer 1 Report
Here are the responses to the authors corrections proposed in the revised manuscript (highlighted yellow).
Thank you for giving me the opportunity to review the manuscript titled: “Implementation of the obturator nerve block into a supra-inguinal fascia iliaca compartment block based analgesia protocol for hip arthroscopy: Retrospective Pre-Post study” written by Seounghun Lee et al.
Point 1: The submission addresses a very interesting topic of anesthesia protocol suitable for hip arthroscopy. The missing information is the pain threshold influence on the fentanyl consumption
We have modified the manuscript as adviced.
‘It is our routine to explain the NRS (0, no pain; 10, worst pain), when PCA should be use (NRS ≥ 4 or when patient feels pain) and how to use the PCA device while filling out the consent form for anesthesia on the day before surgery’
Correction acceptable.
The authors must explain for what test thirteen patients per group have a power of 80% with an alpha of 0.016. Which power sample calculator was used to calculate this result? Supplied explanation by the authors is not sufficient.
When the power sample size was calculated? Before or after the study?
Authors compared 3 groups, but the calculations were made for difference between two independent means by t-test. The explanation is demanded.
We thank the reviewer for the comment. We have modified the manuscript as adviced.
‘Based on difference between two independent means by t-test, thirteen patients per group were required to have a power of 80% with alpha of 0.016 (0.05/3) for post hoc analysis when three groups are significant.’
Correction insufficient .
BY definition the ANOVA testing requires normal distribution.
There is no proof in the tables or text showing the normal distribution of variables.
Yes. You are right. Our main result showed by median [IQR]. The groups were compared using the Kruskal-Wallis test according to Shapiro Wilk-test. We described in the statistical session.
Correction insufficient .
The Shapiro-Wilk test is a way to tell if a random sample comes from a normal distribution. The test gives a W value; small values indicate the sample is not normally distributed.
The Kruskal–Wallis test by ranks or Kruskal–Wallis H test is a non-parametric method for testing whether samples originate from the same distribution.
It is used for comparing two or more independent samples of equal or different sample sizes.
The authors should clearly describe the statistical analysis performed in the study.
There is a huge difference when the Pain score is assessed in a range of 3 hours. The authors wrote that assessment was performed every 3~6 hours in a numeric rating scale (NRS) during the stay in the post-anesthesia care unit and ward.
We have modified the manuscript as follow.
‘Numeric rating scale (NRS) pain score was assessed in post-anesthesia care unit and in ward by nurse regular rounding and patient request for pain control.’
So, we expressed only highest and lowest NRS during postoperative 24 hours.
Correction conditionally acceptable.
This change should appear in the results.
The most important questions concerning the study are focusing on the ethical issues. Please respond to mentioned below questions.
The authors applied different protocols for anesthesia. How did they keep the randomization protocol for this study?
What kind of informed consent was signed by patients?
How much the study is biased while choosing SI-FICB with or without an obturator nerve block?
How the informed consent could be waived if the patients were divided into three groups: group N (no blockade), group F (SI-FICB only), and group FO (SI-FICB with obturator nerve block). How the protocol was presented to IRB and how the approval was obtained?
I am aware the study is retrospective. The patient had to undertake the decision to agree or disagree the proposed anesthesia and treatment.
Was the patient aware about other options: no blockade, SI-FICB only, SI-FICB with obturator nerve block? Was the patient allowed to make / take part in the decision?
I must repeat the question. How the treatment /anesthesia protocol was presented to the patient? Did patient have any chance to make a choice what kind of anesthesia he or she might have?
As stated in the title, this is not RCT. We restrospectively collected and analyzed the data.
The background of our hip arthroscopy postoperative analgesia protocol changes is detailed in the 2. 1 Implementation of subpectineal obturator nerve block.
The fentanyl use in the postoperative period should be clearly described how its consumption changes, and sources and types of influences. The pharmacokinetics should be considered and discussed.
I think the meaning is that it was possible to reduce opioid consumption in FO group.
I'm really sorry, but if the question is not answered properly, please provide me with directions to supplement it in more detail.
Directions are provided in the questions above highlighted yellow

Author Response
We would like to thank the reviewers and editor-in-chief for giving us a chance to revise again. We are very sorry for the lack of sufficient explanation and correction in the last revision. It was modified as follows.
The authors must explain for what test thirteen patients per group have a power of 80% with an alpha of 0.016. Which power sample calculator was used to calculate this result?
Supplied explanation by the authors is not sufficient.
When the power sample size was calculated? Before or after the study?
Authors compared 3 groups, but the calculations were made for difference between two independent means by t-test. The explanation is demanded.
What we want to know is whether the FO group shows a significant result compared to the N group. If the results of the three groups were significant, the sample size was calculated under the condition of alpha of 0.016 (0.05 / 3) as a post hoc test.
The Shapiro-Wilk test is a way to tell if a random sample comes from a normal distribution. The test gives a W value; small values indicate the sample is not normally distributed.
The Kruskal–Wallis test by ranks or Kruskal–Wallis H test is a non-parametric method for testing whether samples originate from the same distribution.
It is used for comparing two or more independent samples of equal or different sample sizes.
The authors should clearly describe the statistical analysis performed in the study
We totally agree with you. We confirmed normality by Shapiro-Wilk test, and since our data was not a normal distribution, we conducted Kruskal–Wallis test and then Dunn’s test by post-hoc test. Although it is a retrospective study, we conducted priori calculation of the minimum size of samples to see whether the limited data already constructed had sufficient power to support the study's results. Since the information given to us in the research preparation phase was mean ± sd, we conducted t-test to estimate the sample size. After the data was obtained, the Kruskal Wallis test was performed because our data was not a normal distribution, and the two groups were compared by Dunn's test. All of these are described in detail in the statistical section.
We are very sorry for the confusion. The part related to sample size was modified as follows.
‘The study sample size was chosen as all patients underwent hip arthroscopy patients from January 2017 to August 2019. All available patients were considered, and priori power analysis was conducted. According to our previous unpublished data, the cumulative consumption of fentanyl at 24 hours after hip arthroscopy was about 700 µg (sd 150). We determined that the 30% reduction of fentanyl consumption has a clinical effect (effect size 1.4). Based on difference between two independent means by t-test, thirteen patients per group were required to have a power of 80% with alpha of 0.016 (0.05/3) for post hoc analysis when three groups are significant. Considering the amount of annual cases (30 cases average), and the ratio of utilizing peripheral nerve block based multimodal analgesia (about 1:1 ratio), it was expected to be more than 13 per group.’
**figure in word file
The most important questions concerning the study are focusing on the ethical issues. Please respond to mentioned below questions.
The authors applied different protocols for anesthesia. How did they keep the randomization protocol for this study?
We did not select sample randomly. As you know, this study is a retrospective design and we analyzed the pain outcome differences caused by changes in the researchers' analgesia protocol during the period. We are sorry for the lack of explanation for N group. We added the following sentence in section 2.1. Implementation of subpectineal obturator nerve block.
‘SI-FICB for hip arthroscopy have been performed by a single anesthesiologist (B.H) since 2017 in our institution.’
‘Patients who have not had any nerve blocks during the same period were assigned to the group N as control group. Thus, the patients were divided into three groups: group N (no blockade), group F (SI-FICB only), and group FO (SI-FICB with obturator nerve block) (Figure 1).’
What kind of informed consent was signed by patients?
The informed consent for the study was waived because it is a retrospective design. When each patient underwent anesthesia, informed consent for anesthesia was obtained. At this time, the explanation of multimodal analgesia including nerve block method and usage of PCA are explained and agreed. This, like all hospitals, is our routine practice.
How much the study is biased while choosing SI-FICB with or without an obturator nerve block?
This study is not RCT and there is a some bias because the two groups that have undergone nerve block divided based on a specific time point. This is explained in the limitations of the study as follows.
‘First, there may have been some unrevealed confoundings due to the retrospective design.’
We tried to reduce the bias by comparing the N groups of the same period and the same surgery performed by the same surgeon.
How the informed consent could be waived if the patients were divided into three groups: group N (no blockade), group F (SI-FICB only), and group FO (SI-FICB with obturator nerve block). How the protocol was presented to IRB and how the approval was obtained?
We retrospectively grouped patients who had already undergone surgery. It was approved by IRB as a retrospective study using medical records.
I am aware the study is retrospective. The patient had to undertake the decision to agree or disagree the proposed anesthesia and treatment.
All patients complete an informed consent for anesthesia before surgery. At this time, the explanation of multimodal analgesia including nerve block method and usage of PCA were explained and agreed.
Was the patient aware about other options: no blockade, SI-FICB only, SI-FICB with obturator nerve block? Was the patient allowed to make / take part in the decision?
The anesthesia method with or without nerve blocks were determined by the anesthesiologist who performed the anesthesia. It was decided by an anesthesiologist in charge of duty on the day of surgery due to differences in interest in regional anesthesia or differences in skill of the procedure. Patients underwent anesthesia in the best way that each anesthesiologist could provide for each period. As described above, the nerve block has been implemented by one anesthesiologist, and the protocol was changed for better analgesic effect during the audit. About half of the patients (47 people) during the same period underwent anesthesia by other anesthesiologists and did not have the chance of nerve block.
I must repeat the question. How the treatment /anesthesia protocol was presented to the patient? Did patient have any chance to make a choice what kind of anesthesia he or she might have?
It was explained during complete an informed consent for anesthesia, the patient usually chooses the pain control method recommended by the anesthesiologist, but they has the right to refuse them.
The fentanyl use in the postoperative period should be clearly described how its consumption changes, and sources and types of influences. The pharmacokinetics should be considered and discussed.
Studies about postoperative pain control usually compare the amount of opioid consumption or pain score after surgery. Significantly less opioid consumption compared to other groups indicates that it is effective in postoperative pain control. The consumption of opioid was obtained from the data of patient controlled analgesia (PCA) devices stored in electronic medial record system. Simply, less opioid consumption means patient uses less PCA device. In this study, it is shown that the obturator nerve block implemented FICB is helpful in controlling pain after hip arthroscopic surgery.
